# The Impact and Efficacy of Vitamin D Fortification

**DOI:** 10.3390/nu16244322

**Published:** 2024-12-14

**Authors:** Ruyuf Y. Alnafisah, Atheer S. Alragea, Mona K. Alzamil, Amani S. Alqahtani

**Affiliations:** Saudi Food and Drug Authority, Riyadh 13513, Saudi Arabia; asraqea@sfda.gov.sa (A.S.A.); mkzamil@sfda.gov.sa (M.K.A.); as.qahtani@sfda.gov.sa (A.S.A.)

**Keywords:** vitamin D, food fortification, fortification doses, fortification methods, stability, bio-accessibility, bioavailability, cost-effectiveness

## Abstract

Vitamin D deficiency is a global health issue linked to various chronic diseases and overall mortality. It primarily arises from insufficient sunlight exposure, compounded by dietary limitations. Vitamin D fortification of commonly consumed foods has emerged as a viable public health intervention to address this deficiency. This review evaluates the impact of vitamin D food fortification on serum levels, intake, and health outcomes and explores the stability, bio-accessibility, bioavailability, and cost-effectiveness of such interventions. A comprehensive literature search was conducted in PubMed and Google Scholar, focusing on studies from 2015 to 2024. The criteria included primary research on healthy adults that addressed the effects of vitamin D fortification on health, intake, and serum levels, as well as the fortification’s stability, bio-accessibility, bioavailability, and cost-effectiveness. Studies were extracted and analyzed according to PRISMA guidelines. The review included 31 studies from diverse geographic locations, revealing that fortifying dairy products, cereals, fats, oils, and other food items effectively increased serum 25-hydroxyvitamin D levels. The fortification methods varied, with vitamin D3 showing superior efficacy over vitamin D2. Encapsulation techniques improved stability and bioavailability. Fortifying staple foods like milk and eggs proved cost-effective compared with pharmaceutical interventions. Vitamin D food fortification significantly enhances serum levels and intake, with dairy and cereals being the most frequently fortified. Standardized fortification guidelines are essential to ensure safety and efficacy. Ongoing evaluation and region-specific policies are crucial for effectively optimizing fortification strategies and addressing vitamin D deficiency.

## 1. Introduction

Epidemiological studies have firmly established a connection between vitamin D and various aspects of health [1,2,3]. Vitamin D plays a crucial role in maintaining musculoskeletal health by regulating calcium and phosphorus levels [4]. Additionally, it contributes to innate immunity and immune-related disorders [5]. Its deficiency is a global health concern, with research spanning basic science to clinical applications consistently demonstrating strong associations with chronic diseases and acute conditions [6,7,8,9,10,11,12,13,14,15]. Moreover, vitamin D deficiency has been linked to overall mortality according to several studies [16,17,18].

Vitamin D is distinct from other nutrients in that it functions more as a prohormone than a typical vitamin. Its primary source is the skin’s exposure to ultraviolet (UV) rays, rather than being obtained solely through dietary intake. Nevertheless, there are two forms of vitamin D found naturally in small quantities within specific foods. These include ergocalciferol (vitamin D2), derived from plant or fungal sources like mushrooms, and cholecalciferol (D3), obtained from animal-based foods such as oily fish, butter, eggs, and liver [19,20,21]. The vitamin D content in these food sources can vary significantly between countries due to environmental and agricultural factors [22].

The consumption of vitamin-D-rich foods varies widely among populations [22]. In Saudi Arabia, for instance, there is a substantial discrepancy between the recommended vitamin D intake levels and the high prevalence of individuals failing to meet these requirements [23,24]. To address this issue, fortifying commonly consumed foods with vitamin D has gained traction as a promising intervention to improve intake levels and combat vitamin D deficiency [25,26,27,28]. However, the success of fortification programs relies on developing and implementing evidence-based policies tailored to the distinct dietary patterns and cultural backgrounds of a region.

The existing literature on food fortification for addressing vitamin D deficiency in Saudi Arabia exhibits notable gaps due to limitations in data sources. Consequently, this review investigated the impact of vitamin D food fortification on health outcomes, serum vitamin D levels, and intake among apparently healthy populations from a global perspective. It also examined the stability, bio-accessibility, bioavailability, and cost-effectiveness of vitamin D fortification.

## 2. Magnitude of Vitamin D Deficiency

Vitamin D deficiency has varying prevalence rates across different regions and demographic groups [29,30,31,32,33,34,35,36,37,38], affecting approximately 1 billion people worldwide [30]. A recent systematic review assessed the prevalence of vitamin D deficiency globally among individuals aged 1 year and older, revealing that 15.7%, 47.9%, and 76.6% of participants exhibited serum 25-hydroxyvitamin D (25-OH D) levels below 30, 50, and 75 nmol/L, respectively [31].

Recent reviews show that the poorest vitamin D status can generally be observed in the Middle East. The median or mean 25-OH D serum level in most surveys is between 25 and 50 nmol/L [32]. Previous systematic reviews also found that the prevalence of vitamin D deficiency in the Middle East varies between 30 and 90% depending on the type of study, country, age group, and assay used [33].

Within the region, the Gulf Cooperation Council (GCC) countries report some of the highest prevalence rates of vitamin D deficiency, characterized by a 25-OH D serum level below 50 nmol/L. The rates observed are as high as 81% among Saudis, 86% among adult Emiratis, 86.4% among healthy adult Bahrainis, 64% among adult Qataris, 100% among Omani women of reproductive age, and over 80% among adult Kuwaitis [34,35,36,37,38].

Given that sunlight exposure accounts for over 90% of vitamin D production in humans [39], vitamin D deficiency is prevalent in Saudi Arabia due to factors such as conservative clothing practices and limited sun exposure [40,41,42]. A series of studies conducted in the central region of Saudi Arabia from 2008 to 2017 examined the prevalence of vitamin D deficiency. Participants of all ages were recruited from various groups. The findings revealed that, over the 10-year period, vitamin D deficiency was prevalent in 73.2% of the population [43]. Additionally, a recent study investigating micronutrient levels in Saudi Arabia found that vitamin D deficiency was the most prevalent, impacting 64.3% of the population [44].

## 3. Public Health Intervention Measures

### 3.1. Global Initiatives

Various regulatory bodies around the world have addressed vitamin D fortification programs. In 1996, the Food and Agriculture Organization (FAO) published a report on food fortification, focusing on technology and quality control. The report highlighted the necessity of developing a reliable, accurate, and rapid methodology for assessing vitamin D status to make required adjustments to fortification practices [45].

Although the World Health Organization (WHO) has no specific recommendations, in 2006, it published guidelines that provided a comprehensive framework for countries considering implementing fortification programs to address micronutrient deficiencies. It emphasizes the importance of understanding the dietary vitamin D intake in the population, considering sunlight exposure, ensuring safe fortification levels to avoid potential toxicity, and tracking the effectiveness of any implemented fortification program [46].

In the United States, the Food and Drug Administration (FDA) has specified which foods can be fortified with vitamin D and the permissible levels. A notable change in July 2016 allowed for an increase in the amount of vitamin D that can be added to milk and authorized the fortification of plant-based milk alternatives (such as soy, almond, and coconut) and plant-based yogurt alternatives. The updated regulations permit up to 84 IU/100 g of vitamin D3 in milk, 84 IU/100 g of vitamin D2 in plant-based milk alternatives, and 89 IU/100 g of vitamin D2 in plant-based yogurt alternatives [47].

In Finland, the Ministry of Agriculture and Forestry enacted a regulation in 2021 that mandates the fortification of skimmed homogenized milk with vitamin D3 [48]. The government of Canada has established a vitamin D strategy, launched in 2022, aimed at significantly increasing vitamin D intake through enhanced food fortification. New regulations have nearly doubled the mandatory vitamin D content in cow’s milk and margarine, with compliance required by 31 December 2025. These regulations also allow for increased vitamin D levels in fortified plant-based beverages, and similar measures are being considered for yogurt [49].

In 2023, the European Food Safety Authority (EFSA) revised the tolerable upper intake level (UL) for vitamin D following requests from the European Commission. The panel established a UL of 100 μg of vitamin D equivalents (VDE) per day for adults. It determined that most European populations are unlikely to exceed this UL, except those regularly consuming high-dose vitamin D supplements [50]. During the same year, Food Standards Australia New Zealand (FSANZ) mandated the fortification of edible oil spreads (such as margarine) with vitamin D to address significant public health concerns in Australia [51].

### 3.2. Regional Initiatives

In 2019, the WHO Eastern Mediterranean Region (EMRO) published a report on wheat flour fortification in Eastern Mediterranean countries. The report highlighted that fortifying industrially processed wheat flour is a cost-effective and efficient way to improve micronutrient intake. While most countries in the region (17 out of 22) have implemented some level of wheat flour fortification, primarily with iron and folic acid, fortification with vitamin D, vitamin A, and zinc is less common (2 out of 22). The report suggests that, despite the benefits, vitamin D fortification has not been widely adopted in the region [52].

### 3.3. Local Initiatives

In Saudi Arabia, the Saudi Food and Drug Authority (SFDA) issued a regulation in 2022 mandating the addition of essential vitamins and minerals to certain food products. This regulation specifically includes guidelines for fortifying commonly consumed items such as milk, dairy products, edible oils, and bread with vitamin D to enhance public health [53].

## 4. Materials and Methods

### 4.1. Literature Search Strategy

A search of an electronic database (PubMed) was conducted over the last 10 years (from 2015 to 2024). To initiate this search, we used a combination of MeSH terms and text words including (“Vitamin D” OR “Cholecalciferol” OR “Hydroxycholecalciferols” OR “Calcifediol” OR “Dihydroxycholecalciferols” OR “24,25-Dihydroxyvitamin D3” OR “Ergocalciferols” OR “25-Hydroxyvitamin D2” OR “Dihydrotachysterol”) AND (“Food fortification” OR “Biofortification” OR “Food Enrichment” OR “Food supplementation” OR “Foods, Fortified” OR “Fortified Food” OR “Fortified Foods” OR “Food, Supplemented” OR “Foods, Supplemented” OR “Supplemented Food” OR “Supplemented Foods” OR “Enriched Food” OR “Enriched Foods” OR “Food, Enriched” OR “Foods, Enriched”). To identify additional potentially relevant articles, Google Scholar was searched, along with a manual search reviewing the reference lists of included studies.

### 4.2. Study Selection

Three independent reviewers were tasked with selecting studies for inclusion in the review, with any disagreements resolved through discussion to achieve consensus. The inclusion criteria specified that only primary research studies were considered, encompassing in vivo, in vitro, clinical trials, surveys, and cohort studies. These studies needed to be published in English and dated between 2015 and 2024. For human studies, participants were required to be apparently healthy adults. Included studies had to address specific outcomes related to the impact of vitamin D food fortification on health, vitamin D intake, or 25-OH D serum levels (including total serum 25-OH D, 25-OH D2, and 25-OH D3), as well as aspects related to the stability, bio-accessibility, bioavailability, and cost-effectiveness of vitamin D food fortification. 

Stability refers to a food system’s ability to maintain adequate nutrient levels despite disturbances [54]. Bio-accessibility is the proportion of a compound released from the food matrix in the gastrointestinal tract that becomes available for absorption [55]. Bioavailability is the rate and extent to which a compound is absorbed and made available for cellular metabolism within the host [55]. Cost-effectiveness involves analyzing the ratio of the cost of a specific intervention relative to its effectiveness [56].

Secondary studies, publications in languages other than English, and research involving populations with particular demographic characteristics, such as individuals with diseases, the elderly, children and adolescents, menopausal women, pregnant or lactating women, and obese individuals, were excluded from this review.

The “preferred reporting items for systematic reviews and meta-analyses (PRISMA)” statement was used to guide and report the methods. 

### 4.3. Data Extraction

Three authors independently extracted data from each study using a data extraction sheet. The extracted data included the year of publication, country, study design, duration, sample size, fortified foods, fortification doses and methods, and key findings related to health impacts, vitamin D intake, 25-OH D serum levels, fortification stability, bio-accessibility, bioavailability, and cost-effectiveness.

## 5. Results

### 5.1. Summary of Included Studies 

#### 5.1.1. Articles General Characteristics

The PRISMA flowchart (Figure 1) shows that 279 titles were screened; of the 45 full-text papers reviewed, 31 articles were included. The studies were conducted across multiple countries, including Iran, Denmark, Slovenia, Australia, Italy, Pakistan, the United States (US), China, the United Kingdom (UK), Germany, the Netherlands, Sweden, Finland, Israel, Belgium, Canada, and Ireland (Table 1). Among the included studies, 27 were conducted on human subjects, 11 were simulation studies [57,58,59,60,61,62,63,64,65,66,67], 15 were randomized controlled trials (RCTs) [68,69,70,71,72,73,74,75,76,77,78,79,80,81,82], and 1 was a cohort study [83]. The remaining four studies involved in vitro or in vivo research [84,85,86], and one was categorized as experimental [87]. The human studies primarily focused on healthy adults, except for five that included participants across different age groups (children, teenagers, and adults) [58,61,62,64,67].

#### 5.1.2. Fortified Foods

The fortified foods examined in the reviewed articles encompassed a diverse selection, including dairy products (such as butter, milk, yogurt, cheese, doogh, and ice cream) [57,58,59,60,61,63,64,65,66,67,69,70,75,76,80,81,83], cereal products (such as white bread, whole bread, ready-to-eat breakfast cereals, biscuits, crisp bread, and wheat flour) [57,58,59,61,62,63,64,67,71,72,74,76,77,80,82,86], fats and oils (such as canola oil, sunflower oil, margarine, mozzarella cheese baked on pizza, corn oil, flaxseed oil, fish oil, and mayonnaise) [58,59,61,64,65,68,73,78,79,81,84,85], vegetables, fruit, and their products (such as iced fruits and orange juice) [58,59,61,64,74,77,82], animal products (such as meat, fish, poultry, and eggs and their products) [60,61,63,70,80], and specific items (such as sugar, honey, beverages, spices and other ingredients, and ricotta cheese) [58,61,87].

#### 5.1.3. Fortification Doses

The fortified doses varied across studies in terms of units and quantity and are detailed in Table 1. Units of measurement included International Units (IU) per day [69,70], IU per specific amount in grams [66,68,73,78,79,85,87], micrograms (μg) per day [63,71,74,77,80,81,82], μg per gram [59,65,67,72,83], and μg per unit of energy (expressed in kilocalories (kcal) or megajoules (MJ)) [61,64]. 

The fortification doses for dairy products showed considerable variability. For instance, milk was fortified at 800 IU/day [69], while various milk and cheese products had doses between 800 and 1200 IU/10 MJ or 239 kcal. Similarly, fortified yogurt ranged from 800 to 1200 IU/day [76,78,80].

Fats and oils were fortified with doses such as 1000 IU/25 g in canola oil [68,78], 500 IU/30 g in sunflower oil [79], 600 and 800 IU/100 g in margarine [65,83], and 200 IU or 28,000 IU/28 g in mozzarella cheese baked on pizza [73]. 

Cereal products received fortification including 168 or 200 IU/100 g in breakfast cereals [59,62], 600 IU/day in biscuits [77,82], and 1000 IU/50 g or 800, 1000, or 1200 IU/day in fortified bread [63,71,72,76,80]. 

Vegetables, fruits, and animal products were fortified with doses including 800, 1000, or 1200 IU µg/10 MJ or 239 kcal [61].

#### 5.1.4. Fortification Processes

Out of the 31 studies reviewed, only 8 provided specific details on fortification methods. These methods included adding vitamin D3 as a concentrated solution in a food-grade emulsifier base [68,73], encapsulating vitamin D3 within reassembled casein micelles along with polysorbate-80 [70], incorporating lyophilized UV-treated yeast [71], adding vitamin D3 during homogenization at 85 °C for 30 min [87], encapsulating vitamin D using protein-based carriers [84,85], and incorporating either lyophilized UV-treated yeast or crystalline vitamin D2 into dough before baking [86].

### 5.2. Summary of the Findings

#### 5.2.1. Impact of Fortification on 25-Hydroxyvitamin D Serum Levels

Prospective studies showed increases in 25-OH D serum levels following vitamin D fortification of dairy products [70,75,76,80,83]; fats and oils [68,73,78,79,83]; cereal products [72,74,76,80,82]; vegetables, fruit, and their products [74,82]; and animal products [76,80]. These studies demonstrate the broad effectiveness of fortification across a variety of food types, with the extent of serum 25-OH D increase varying according to the dose, food matrix, and duration of the intervention.

For instance, one RCT reported a 2-week fortification of yogurt with 50,000 IU (1250 µg) per day, which led to an approximate increase of 8 ng/mL in serum 25-OH D [70]. Bread fortified with 1000 IU (25 µg) per 50 g for 8 weeks resulted in a significant increase of 39 ng/mL (*p* < 0.001) in serum 25-OH D in 30 participants [72].

Two RCTs investigated the fortification of canola oil with 1000 IU (25 µg) per 25 g, resulting in a statistically significant increase in serum 25-OH D levels of 2.85 ± 4.69 ng/mL over 3 months [68,78].

Two studies fortified orange juice and biscuits with 600 IU (15 µg) per day for 12 weeks in 335 participants found a significant increase in serum 25-OH D levels from 53 to 103 nmol/L (*p* < 0.0001) [74] and 15.3 to 16 ng/mL (*p* < 0.001) [82]. 

Other studies assessed the effectiveness of vitamin D fortification using a combination of dairy products (yogurt, cheese, eggs, and crisp bread) to deliver 800 IU (20 µg) and 1200 IU (30 µg) of vitamin D daily over 12 weeks in 143 participants, resulting in a significant increase in serum 25-OH D levels from 10.5 ng/mL to 26.4 ng/mL (*p* < 0.001) [76] and 10.2 ng/mL to 24.5 ng/mL (*p* < 0.05) [80].

A dose-dependent effect was observed, where mozzarella cheese baked on pizza was fortified with either 200 IU (5 µg) or 28,000 IU (700 µg) per 28 g. The high-dose group (n = 49) showed a substantial increase of 73 ± 22 nmol/L (*p* < 0.0001) after 10 weeks, compared to a more modest increase of 5.1 ± 11 nmol/L (*p* = 0.003) in the low-dose group (n = 47) [73].

A long-term cohort study over 11 years also demonstrated positive outcomes, examining the fortification of margarine and milk with 800 IU (20 µg) and 20 IU (0.5 µg) of vitamin D per 100 g, respectively; both food vehicles resulted in a significant increase of 17 nmol/L in serum 25-OH D (*p* < 0.001) [83].

#### 5.2.2. Impact of Fortification on Vitamin D Intake

Simulation studies evaluated fortification levels to optimize daily vitamin D intake from various food groups: dairy products [57,58,59,60,61,63,64,65,66,67]; fats and oils [58,59,61,64,65]; cereal products [57,58,59,61,62,63,64,67]; vegetables, fruit, and their products [58,59,61]; animal products [58,61,63]; and specific items, including sugar, honey, beverages, spices, and other ingredients [58,61].

Pourmohamadkhan et al. in Iran suggested that fortifying foods like butter, milk, and yogurt at 3 μg/100 g would optimize vitamin D intake [57]. Similarly, Sengupta et al. in Denmark found that fortification across various foods, including milk and cereals, could help 70% of the population exceed average vitamin D requirements safely [58]. In Slovenia, a study demonstrated a nearly five-fold increase in vitamin D intake with fortification of eggs, milk, and yogurt [60].

Other studies, such as Grønborg et al. in Denmark, found that fortifying yogurt and eggs with 800 IU/day was sufficient to achieve safe intake levels [63], while Moyersoen et al. showed that fortification of bread, cereals, and dairy with 276 IU/100 kcal reduced vitamin D inadequacy from 92–96% to less than 2% [64]. Harika et al. in multiple countries confirmed that fortifying margarine and milk with 600 IU/100 g and 40 IU/100 mL, respectively, could double vitamin D intake [65].

Fortifying milk and yogurt allowed 80% of the population to meet recommended levels [66], while fortifying wheat flour and milk in the UK effectively reduced vitamin D deficiency without exceeding safe limits [67]. In Ireland, a study demonstrated that fortifying milk, cereals, and margarine could significantly increase median intake, from 3 μg/day to 12 μg/day [59].

#### 5.2.3. Impact of Fortification on Health

Three RCTs have investigated the effects of vitamin D fortification on various health biomarkers, including blood pressure (BP), blood lipids, parathyroid hormone (PTH) levels, and indicators of obesity [73,75,79].

One study, conducted by Martucci et al. (2020) in Italy, focused on the fortification of milk. In this trial, 48 participants aged 63–80 years received fortified milk, which resulted in a 29% increase in serum 25-OH D levels. However, the study found no significant impact on blood pressure (BP) or blood lipid profiles [75].

In a separate trial by Nikooyeh et al. (2020) in Iran [79], 65 participants were administered sunflower oil fortified with 500 IU (12.5 µg) of vitamin D per 25 g. This fortification resulted in notable reductions in several obesity-related indicators, including weight, body mass index (BMI), waist girth, as well as reductions in serum cholesterol and low-density lipoprotein (LDL) cholesterol levels. Furthermore, serum intact parathyroid hormone (iPTH) levels decreased by 10.2 pg/mL (*p* = 0.009) [79].

Another trial, led by Al-Khalidi et al. (2015) in Canada, investigated the impact of mozzarella cheese fortified with varying doses of vitamin D, either 200 IU (5 µg) or 28,000 IU (700 µg) per 28 g, in a sample of 96 participants aged 18–70 years. The study demonstrated that the high-dose fortification led to a significant reduction in serum PTH levels [73]. 

#### 5.2.4. Stability, Bio-Accessibility, Bioavailability, and Cost-Effectiveness of Vitamin D Fortification 

Only one study has evaluated the stability of fortified vitamin D, demonstrating that vitamin D remains stable throughout the processing and storage of fortified ricotta cheese, showing high heat stability and consistent distribution [87].

Two studies have investigated the bio-accessibility of vitamin D [85,86]. The first study explored the encapsulation of vitamin D using protein-based carriers to evaluate the bio-accessibility of vitamin D in various oils. Vitamin D3 was dissolved in medium-chain triglycerides at a concentration of 1.0 million IU per gram. Oil-in-water emulsions were prepared with a 90% aqueous phase and a 10% oil phase using flaxseed, corn, or fish oils. Vitamin-D3-enriched oil was incorporated into the oil phase. The emulsions were subjected to in vitro digestion through a simulated gastrointestinal tract (GIT) at 37 °C, with phases that mimicked the conditions of the mouth, stomach, and intestines. The results indicated that vitamin D was significantly more bio-accessible in nano-emulsions containing monounsaturated oils (78%), such as corn oil, compared to those containing polyunsaturated oils (43%), such as flaxseed and fish oils (*p* < 0.05) [85].

The second study explored various methods of fortifying bread. Sandwich breads were prepared in an industrial test kitchen and kept frozen at −20 °C. The breads were fortified either with lyophilized UV-treated yeast or crystalline vitamin D2, aiming for a target fortification level of 100 IU per 50 g serving of fresh bread after baking. These fortified breads were subjected to simulated digestion mimicking the oral, gastric, and intestinal phases. The results showed that bread fortified with crystalline vitamin D2 had significantly higher bio-accessibility (70.7–84.8%) compared to bread fortified with UV-treated yeast (6.0–7.5%) [86].

Bioavailability was assessed across seven studies, examining various forms and fortification methods [70,71,74,77,81,82,84]. For example, Itkonen et al. demonstrated that women consuming bread fortified with 1000 IU of vitamin D2 per day showed a significant increase in serum 25-OH D2 levels, but there was no significant effect on total or 25-OH D3 levels [71]. Multiple studies reported that orange juice and biscuits fortified with 600 IU of vitamin D3 daily led to a significant increase in serum 25-OH D, with the D3 group showing a greater rise compared to the D2 group (*p* < 0.0001) [74,77,82]. Additionally, it was observed that 800 IU of vitamin-D3-fortified milk was 1.5 times more effective than vitamin D2 at increasing postprandial serum 25-OH D levels [81].

In a study by Levinson et al. [70], 87 participants aged 18–61 years were given yogurt fortified with 50,000 IU of vitamin D3 per day, using encapsulation techniques involving rCMs and polysorbate-80. The results revealed a significant increase in serum 25-OH D levels by approximately 8 ng/mL. In another study, Khan et al. [84] investigated the bioavailability of vitamin D encapsulated in mayonnaise, utilizing whey and soy proteins as carriers in a rat model. The study found that both encapsulation techniques significantly enhanced the bioavailability and stability of vitamin D. Specifically, rats in the intervention group, which consumed the vitamin-D-fortified mayonnaise, exhibited significantly higher serum 25-OH D levels (58.14 ± 6.29 nmol/L) compared to the control group (37.80 ± 4.98 nmol/L).

One study evaluated the cost-effectiveness of vitamin D fortification, demonstrating that fortifying milk and eggs with vitamin D is five times more cost-effective than the cheapest prescription drugs [60].

**Table 1 nutrients-16-04322-t001:** Summary of included studies.

Author/Year	Country	Study Design	Study Sample	Fortified Food	Fortification Dose	Fortification Process	Key Findings
Pourmohamadkhan, M., et al., 2023 [57]	Iran	Simulation study	9704 Iranian aged 35–65 years	Butter, white bread, whole bread, milk, yogurt, cheese, and doogh	NA	NA	- Optimal fortification was at 3 μg/100 g
Sengupta, S., et al., 2023 [58]	Denmark	Simulation study	3952 participants aged 4–75 years	- Milk, cheese, cereals, vegetables, fruit, meat, fish, poultry, eggs, fats, oils, and their products. - Ice cream, fruit ice, and other edible ices, sugar, honey, beverages, spices and other ingredients, potato, and juice	NA	NA	- 70% of the population exceeds average vitamin D requirements without exceeding the upper intake level
Vičič, V., et al., 2022 [60]	Slovenia	Simulation study	176 participants aged 44–65 years	Egg, milk, and yogurt	NA	NA	- Fortification increased vitamin D intake 4.8-fold - Fortification increases vitamin D intake nearly five-fold cost-effectively versus prescription drugs
Ghasemifard, N., et al., 2022 [68]	Iran	RCT	93 participants aged 18–30 years	Canola oil	1000 IU (25 µg)/25 g	Direct addition of vitamin D3	- Vitamin D supplements or fortified oil significantly enhance serum 25-OH D levels (*p* = 0.001)
Christensen, T., et al., 2022 [61]	Denmark	Simulation study	3946 participants aged 4–75 years	- Milk, cheese, cereals, vegetables, fruit, meat, fish, poultry, eggs, fats, oils, and their products. - Ice cream, fruit ice, and other edible ices, sugar, honey, beverages, spices and other ingredients, potato, and juice	800. 1000, or 1200 IU (20, 25, or 30 µg)/10 MJ (239 kcal)	NA	- The minimum effective fortification level was 12 µg/10 MJ.
Foulkes, S., et al., 2021 [69]	Australia	RCT	180 participants aged 50–79 years	Milk	800 IU (20 µg)/day	NA	- Fortification had no effect on BP or blood lipids
Martucci, M., 2020 [75]	Italy	RCT	48 participants aged 63–80 years	Milk	NA	NA	- Fortified milk increased serum 25-OH D levels by 29%
Grønborg, I. M., et al., 2020 [76]	Denmark	RCT	143 Danish and Pakistani females aged 18–50 years	Yogurt, cheese, eggs, and crisp bread	800 IU (20 µg)/day	NA	- Fortification effectively increased serum 25-OH D levels (*p* < 0.01)
Calame, W., et al., 2020 [62]	UK	Simulation study	3770 participants aged 4 years and older	Ready-to-eat breakfast cereals	168 IU (4.2 μg)/100 g	NA	- Approximately 34 g is needed for adults to reach 50% of the daily vitamin D intake
Durrant, L. R., et al., 2020 [77]	UK	RCT	335 White European and South Asian females aged 18–50 years	Orange juice and biscuits	600 IU (15 µg)/day	NA	- Vitamin D3 was more effective than vitamin D2 at raising serum 25-OH D levels
Ghasemifard, N., et al., 2020 [78]	Iran	RCT	93 participants aged 18–30 years	Canola oil	1000 IU (25 µg)/25 g	NA	- In the vitamin-D-sufficient subgroup, serum 25-OH D levels increased more with fortified oil compared to controls (*p* = 0.001).
Nikooyeh, B., et al., 2020 [79]	Iran	RCT	65 participants, mean age 32.5 ± 4 years	Sunflower oil	500 IU (12.5 µg)/25 g	NA	- Fortification increases serum 25-OH D levels (8.8 ng/mL, *p* < 0.001) and decreases intact parathyroid hormone levels (−10.2 pg/mL, *p* = 0.009). - The intervention group showed significant reductions in weight, BMI, waist girth, total cholesterol, and LDL-C compared to controls.
Grønborg, I. M., et al., 2019 [63]	Denmark	Simulation study	855 participants aged 18–50 years	Yogurt, cheese, eggs, and crisp-bread	800 IU (20 µg)/day	NA	- Safe intake levels achieved with yogurt, cheese, eggs, and crispbread providing 20 µg/day.
Grønborg, I. M., et al., 2019 [80]	Denmark	RCT	143 Danish and Pakistani females aged 18–50 years	Yogurt, cheese, eggs, and crisp-bread.	1200 IU (30 μg)/day	NA	- Significant increase in serum 25-OH D levels in fortified groups (*p* < 0.05).
Moyersoen, I., et al., 2019 [64]	Belgium	Simulation study	3200 participants aged 3–64 years	Bread, breakfast cereals, fats and oils, fruit juices, milk, and yogurt and cream cheese	276 IU (6.9 μg)/100 kcal	NA	- Fortification reduced inadequate vitamin D intake from 92–96% to under 2%.
Lovegrove, J. A. 2017 [81]	UK	RCT	17 men with suboptimal vitamin D status, aged 30–65 years	Milk	800 IU (20 μg)/day	NA	- Vitamin-D3-fortified milk was 1.5 times more effective than vitamin D2 at raising postprandial serum 25-OH D levels
Jääskeläinen, T., et al., 2017 [83]	Finland	Cohort Study	17,251 Finnish participants, aged 30 years and older	- Margarine - Milk	- 800 IU (20 µg)/100 g - 20 IU (0.5 µg)/100 g	NA	- Fortification increases serum 25-OH D levels from 48 nmol/L in 2000 to 65 nmol/L in 2011
Harika, R. K., et al., 2017 [65]	Netherlands, the UK, and Sweden.	Simulation study	3411 participants aged 18–50 years	- Margarine - Milk	- 600 IU (15 μg)/100 g - 40 IU (1 μg)/100 mL	NA	- Adherence to fortification guidelines could double vitamin D intake.
Tripkovic, L., et al., 2017 [82]	United Kingdom	RCT	335 participants aged 18–61 years	Juice and biscuits	600 IU (15 μg)/day	NA	- Vitamin D3 was more effective in improving serum 25-OH D levels than vitamin D2
Levinson, Y., et al., 2016 [70]	Israel	RCT	87 participants aged 18–61 years	Yogurt	50,000 IU (1250 μg)/day	Encapsulation techniques (rCMs and polysorbate-80)	- Both techniques raised serum 25-OH D levels by about 8 ng/mL
Itkonen, S. T., et al., 2016 [71]	Finland	RCT	33 female aged 20–37 years	Bread	1000 IU (25 µg)/day	UV-treated yeast	- Vitamin D2 bread significantly increased serum 25-OH D2 level but did not affect total or serum 25-OH D3 levels.
Nikooyeh, B., et al., 2016 [72]	Iran	RCT	90 participants aged 20–60 years	Bread	1000 IU (25 µg)/50 g	NA	- Both fortified bread and vitamin D supplements significantly increased serum 25-OH D levels (*p* < 0.001), with no significant difference between them.
Ejtahed, H. S., et al., 2016 [66]	Iran	Simulation study	5224 Iranian participants aged 18–50 years	- Milk - Yogurt	- 2 IU (0.5 µg)/100 g - 89 IU (2.2 µg)/100 g	NA	- About 80% of the population meet the recommended intake without exceeding the upper intake level.
Al-Khalidi, B., et al., 2015 [73]	Canada	RCT	96 participants aged 18–70 years	Mozzarella cheese baked on pizza	200 IU (5 µg) or 28,000 (700 µg) IU/28 g	Direct addition of vitamin D3	- Low-dose vitamin D increased serum 25-OH D levels by 5.1 nmol/L, while high-dose increased it by 73 nmol/L and significantly reduced serum PTH levels.
Allen, R. E., et al., 2015 [67]	UK	Simulation study	2127 participants aged 8 months and older	- Wheat flour for domestic use, wheat-flour-containing food except for noodles and pasta. - Milk, milk-containing foods except for cream, cheese, and yogurt	- 200 to 1200 IU (5 to 30 μg)/100 g - 20 to 280 IU (0.5 to 7 μg)/100 L	NA	- Fortifying wheat flour with 10 μg/100 g effectively reduced the proportion of at-risk groups without exceeding the upper intake level.
Tripkovic, L., et al., 2015 [74]	Australia	RCT	335 women aged 20–64	Orange juice and biscuit	600 IU (15 μg)/d	NA	- Fortification significantly increased serum 25-OH D levels, with the D3 group showing a greater increase compared to the D2 group (*p* < 0.0001).
Black, L. J., et al., 2015 [59]	Ireland	Simulation study	2653 participants aged 18–64 years	- Milk and its alternatives, yogurt, cream, cheese, fruit juice/drinks, and bread/rolls. - Ready-to-eat breakfast cereals - Margarine	- 80 IU (2 μg)/100 mL - 200 IU (5 μg)/100 g - 320 IU (8 μg)/100 g	NA	- Fortification doses from various foods increase median intake from 3 μg/d to 7 μg/d, 9 μg/d, and 12 μg/d, respectively.
Nzekoue, F. K., et al., 2021 [87]	Italy	Experimental Study	NA	Ricotta cheese	2000 IU (50 µg)/100 g	Direct addition of vitamin D3	- Vitamin D distributes evenly in ricotta cheese after homogenization, and vitamin D3 maintains high heat stability (93.8 ± 1.8%) throughout the food’s shelf life.
Khan, W. A., et al., 2020 [84]	Pakistan, US, and China	Vitro and Vivo study	32 Sprague Dawley rats	Mayonnaise	NA	Encapsulation techniques (whey and soy protein)	- Vitamin-D-fortified mayonnaise effectively increased serum calcium levels in vitamin-D-deficient rats. - Both whey and soy protein encapsulate enhance vitamin D bioavailability and stability.
Schoener, A. L., et al., 2019 [85]	Germany and US	Vitro study	NA	Corn oil, flaxseed oil, and fish oil	1.0 million IU (25,000 µg)/g	Encapsulation techniques	- Vitamin D3 bio-accessibility is higher in nanoemulsions with monounsaturated oils, like corn oil, than in those with polyunsaturated oils, like flaxseed or fish oil.
Lipkie, T. E., et al., 2016 [86]	US	Vitro study	NA	Bread	100 IU (2.5 µg)/50 g	UV-treated yeast or crystalline vitamin D2 in dough	- Bio-accessibility from crystalline vitamin-D2-fortified bread was about 4 times higher than from yeast-fortified bread

NA: not available/applicable. RCT: randomized controlled trials.

## 6. Discussion

Given the growing recognition of the importance of addressing 25-OH D serum level deficiency, this review assessed the impact of such fortification on health outcomes, 25-OH D serum levels, and vitamin D intake among apparently healthy adults. Additionally, we sought to explore the stability, bio-accessibility, bioavailability, and cost-effectiveness of vitamin D fortification. This investigation was particularly pertinent as part of the SFDA’s responsibilities to update current policies by focusing on exploring related aspects, including the types of foods to be fortified, effective fortification techniques, and optimal fortification levels. Given the potential gaps in the current literature, especially in the context of Saudi Arabia, we also addressed these gaps and highlighted the need for evidence-based, region-specific fortification policies.

The current review highlights a diverse range of foods that have been investigated for vitamin D fortification, including dairy products, fats and oils, cereal products, vegetables, fruits, animal products, and specific items like sugar and beverages. We demonstrated that fortifying these commonly consumed foods with vitamin D effectively increases 25-OH D serum levels [68,70,72,73,74,75,76,78,79,80,82,83]. This evidence supports the efficacy of fortification as a viable strategy for mitigating vitamin D deficiency [6,88].

This review also showed that dairy and cereal products are commonly targeted for fortification [57,58,59,60,61,63,64,65,66,67,68,69,70,73,75,76,78,79,80,81,83,84,85]. This might be due to their stability, affordability, and widespread consumption [89,90,91,92,93]. Indeed, fortifying staple foods, like milk, with vitamin D proved to be a more cost-effective measure than pharmaceutical interventions [60], highlighting its economic feasibility and potential for broad public health impact.

On the other hand, fortification of less conventional items, such as certain oils and specific ingredients, was observed in several studies [58,59,60,61,63,64,65,68,70,73,74,77,78,79,80,81,82,84,85,87]. The diverse choices for fortification underscore the need for tailored approaches based on regional dietary patterns and consumption habits. 

This review shows that the amount of vitamin D fortification varies significantly across studies and food products. These inconsistencies underscore the need for standardized guidelines to ensure that fortification levels are both effective and safe.

This review highlights several effective fortification methods, including directly adding concentrated vitamin D3 solutions [68,73,87], one of the most commonly used approaches [94,95,96]. However, this method has been associated with vitamin D instability within the food matrix [97].

Encapsulation techniques have also been utilized to fortify food with vitamin D [70,84,85]. These techniques are designed to address stability concerns [97]. They involve encasing the bioactive core material with secondary wall substances that shield it from external conditions [98,99,100,101]. Besides protecting the bioactive compound, these methods facilitate the controlled release of the encapsulated ingredient and enhance its physicochemical stability [97].

To fortify bread with vitamin D, two studies utilized yeast exposed to UV light [71,86]. UV-treated yeast provides a natural source of this vitamin, particularly for baked products. Since yeasts naturally produce ergosterol, producing vitamin D2 in these organisms only requires UV exposure [102]. Although the UV treatment of yeast for vitamin D2 production is receiving growing attention from both the scientific and industrial communities, there is some debate about the bioavailability of vitamin D2 in the resulting bread [83,102,103].

Vitamin D3 was found to be more effective than vitamin D2 in increasing 25-OH D serum levels [71,74,77,81,82]. This enhanced efficacy can be attributed to chemical differences between the two vitamins. Although their structures are similar, vitamin D2 contains an additional double bond and a methyl group, contributing to its reduced stability. This instability may impair vitamin D2’s effectiveness as a substrate at various stages of the vitamin D metabolism pathway [104]. However, a large-scale, well-designed randomized controlled trial is still required to provide conclusive evidence of this superiority [104].

This study offers a comprehensive review of various food products and vitamin D fortification techniques, providing valuable insights into their effectiveness and feasibility. By examining diverse study types, this review strengthened its conclusions, focusing on healthy adults to ensure its findings are relevant to public health interventions aimed at preventing 25-OH D serum level deficiency. Key outcomes, such as vitamin D stability, bioavailability, and cost-effectiveness, were explored. However, including studies with varied methods and populations introduced a level of heterogeneity. Additionally, most studies focus on short-term outcomes, offering limited insights into the long-term effects of vitamin D fortification. 

This study highlights the need for further research on optimal fortification doses, the comparative efficacy of vitamin D2 versus D3, cultural influences on vitamin D intake, and the cost-effectiveness of fortification strategies. Policymakers are encouraged to implement region-specific vitamin D fortification programs that consider local dietary habits, sun exposure, and cultural practices affecting 25-OH D serum levels. These programs should prioritize fortification methods that enhance the bioavailability and stability of vitamin D, such as encapsulation techniques and the use of vitamin D3 over D2. 

Moreover, governments and health organizations must establish standardized guidelines for fortification doses to ensure both efficacy and safety, preventing risks of overdose or inadequate supplementation. Continuous monitoring and evaluation are essential to ensuring that fortification efforts meet public health objectives. Finally, future studies should investigate how increased serum 25-OH D concentrations may influence health outcomes, particularly among populations at risk of deficiency, such as pregnant women, the elderly, and people with specific health conditions, including cancer, cardiovascular disease, and diabetes mellitus.

## 7. Conclusions

This review assessed the impact of vitamin D food fortification on health outcomes, 25-OH D serum levels, and vitamin D intake in healthy adults while exploring the stability, bioavailability, and cost-effectiveness of fortification. Dairy products and cereals were the most frequently targeted due to their affordability and stability. Fortification methods, including encapsulation techniques, were effective, with vitamin D3 consistently outperforming D2 in elevating 25-OH D serum levels. This review underscores the need for standardized fortification guidelines to maximize safety and efficacy while addressing regional dietary patterns and sunlight exposure. Fortification offers a cost-effective public health strategy, particularly in areas with limited sunlight. Continuous evaluation and tailored policies are crucial to optimizing the impact of fortification programs.

## Figures and Tables

**Figure 1 nutrients-16-04322-f001:**
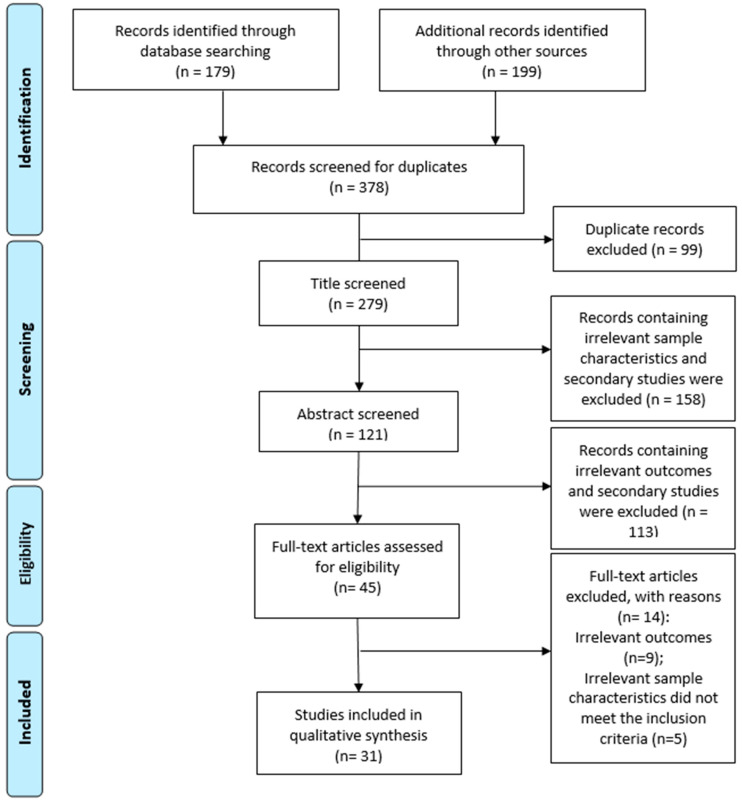
PRISMA flow diagram.

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
