# Peer review of "The Impact and Efficacy of Vitamin D Fortification"

_nutrients, 2024, doi:10.3390/nu16244322_

Round 1

Reviewer 1 Report

Comments and Suggestions for Authors

Dear authors,

I have some suggestions and specific comments to improve the manuscript.

Lines 196-203: Be consistent when listing food categories, some are capitalized, others are capitalized.

Lines 205-217: Please recheck reference no. 44. Namely, you state the dose of vitamin D as 20, 25 and 30 µg/g. In fact, the doses are 20, 25 and 30 mcg/MJ, i.e. per unit of energy and not per gram. The same error is repeated in Table 1.

Table 1: In Table 1, unify the doses, e.g. you use 800 UI/day, and in some places, e.g. 15 µg/d. In some places you use only milk, in others skimmed milk, for this review only milk would probably be sufficient. Then you use dairy drinks and fluid milk products, which are probably the same. Author Durrant (56) has an error in the country record.

Table 1: At reference 62 it is important to use a comma: Both vitamin D from fat-free yogurt, in re-assembled casein micelles (rCMs) and polysorbate-80 (PS80/Tween-80), significantly...

Author Response

Response and Revisions

Thank you for your thoughtful evaluation and for scoring the manuscript positively. We are grateful for your recognition of its significance, organization, scientific rigor, and clarity. Your feedback is invaluable, and we appreciate your time and effort in reviewing our work. Thank you once again for your support.

Comments 1: Lines 196-203: Be consistent when listing food categories, some are capitalized, others are capitalized.

Response 1: Thank you for pointing this out. Adjustments are made as per the comment.

Comments 2: Lines 205-217: Please recheck reference no. 44. Namely, you state the dose of vitamin D as 20, 25 and 30 µg/g. In fact, the doses are 20, 25 and 30 mcg/MJ, i.e. per unit of energy and not per gram. The same error is repeated in Table 1.

Response 2: Thank you for pointing this out. Adjustments are made as per the comment.

Comments 3: Table 1: In Table 1, unify the doses, e.g. you use 800 UI/day, and in some places, e.g. 15 µg/d.

Response 3: Thank you for pointing this out. We appreciate your suggestion regarding the unification of dosing units. However, we believe that preserving the original units as reported in the studies is crucial for accurately representing the variability in fortification practices. Converting these units could lead to potential inaccuracies and misinterpretations of the data. We emphasized the inconsistency in dosing units in the manuscript to provide context for readers. Thank you for your understanding and for your valuable feedback.

Comments 4: In some places you use only milk, in others skimmed milk, for this review only milk would probably be sufficient. Then you use dairy drinks and fluid milk products, which are probably the same. Author Durrant (56) has an error in the country record.

Response 4: Thank you for pointing this out. Adjustments are made as per the comments.

Comments 5: Table 1: At reference 62 it is important to use a comma: Both vitamin D from fat-free yogurt, in re-assembled casein micelles (rCMs) and polysorbate-80 (PS80/Tween-80), significantly....

Response 5: Thank you for pointing this out. Adjustments are made as per the comments

Reviewer 2 Report

Comments and Suggestions for Authors

This is a worthy review topic because the supply of vitamin D in fortified food is a haphazard way of improving vitamin D status of populations, with the uncertainty of population consumption of those foods. There are a few details which could be improved.

1.     A review of the literature on a wide topic is a means of improving the understanding of readers about the topic. However, the very large Table 1, summarising the information in each published paper reviewed, makes it very difficult for the reader to identify the key points of the review. In addition, the very small typeface of the Table makes it extremely difficult to read. This Table really neutralises the purpose of the review. It would be much improved if the key points of the published studies were to be summarised in short, simple tables.

2.     Figure 1 gives the sequence of steps in the selection of publications to be included in this overview. It is not clear what the exclusion criteria were for each step in the selection of studies for review. The reader needs to have confidence that there was no selection bias in excluding publications, so an indication of the selection/exclusion criteria would ensure that the process was objective.

3.     Throughout the text the term “vitamin D levels” in blood serum is used to describe vitamin D status. What was actually measured in serum was not vitamin D but 25-hydroxyvitamin D. Because serum also contains vitamin D as well as 25-hydroxyvitamin D, the term “vitamin D levels” is ambiguous. The text should be corrected to use the term “25-hydroxyvitamin D concentration” or “vitamin D status”.

4.     Line 174: The words here are not well expressed: “Three authors were independently extract the data from each study….”

5.     At lines 188-189, the words “studies involved in vitro or in vivo research” are used to describe the general characteristics of the articles under consideration. It is not clear what the relation is of in vitro studies to the assessment of food fortification on vitamin D status of populations.

6.     Line 333: “vitamin D3 features and additional double bond and a methyl group” This refers to vitamin D2, not vitamin D3.

Comments on the Quality of English Language

Minor corrections to use of English required in some parts of the manuscript.

Author Response

Response and Revisions

Thank you for your thoughtful evaluation of our article. We appreciate your feedback on its significance, organization, scientific soundness, references, and readability. We understand your ranking, particularly your concerns regarding the work's contribution and organization. We took your comments into consideration and have made revisions to enhance the overall clarity and impact of the manuscript.

Comments 1: This is a worthy review topic because the supply of vitamin D in fortified food is a haphazard way of improving vitamin D status of populations, with the uncertainty of population consumption of those foods. There are a few details which could be improved.

  1. A review of the literature on a wide topic is a means of improving the understanding of readers about the topic. However, the very large Table 1, summarising the information in each published paper reviewed, makes it very difficult for the reader to identify the key points of the review. In addition, the very small typeface of the Table makes it extremely difficult to read. This Table really neutralises the purpose of the review. It would be much improved if the key points of the published studies were to be summarised in short, simple tables.
  2. Figure 1 gives the sequence of steps in the selection of publications to be included in this overview. It is not clear what the exclusion criteria were for each step in the selection of studies for review. The reader needs to have confidence that there was no selection bias in excluding publications, so an indication of the selection/exclusion criteria would ensure that the process was objective.
  3. Throughout the text the term “vitamin D levels” in blood serum is used to describe vitamin D status. What was actually measured in serum was not vitamin D but 25-hydroxyvitamin D. Because serum also contains vitamin D as well as 25-hydroxyvitamin D, the term “vitamin D levels” is ambiguous. The text should be corrected to use the term “25-hydroxyvitamin D concentration” or “vitamin D status”.
  4. Line 174: The words here are not well expressed: “Three authors were independently extract the data from each study….”
  5. At lines 188-189, the words “studies involved in vitro or in vivo research” are used to describe the general characteristics of the articles under consideration. It is not clear what the relation is of in vitro studies to the assessment of food fortification on vitamin D status of populations.
  6. Line 333: “vitamin D3 features and additional double bond and a methyl group” This refers to vitamin D2, not vitamin D3.

    Response 1:

    For point 1: Thank you for pointing this out. We agree that the table is very large and it is written in small typeface, which make it difficult to read. Thus, we have condensed it from 10 pages to 5 and standardized the font style to match the rest of the document.

    For point 2: Thank you for your valuable feedback regarding Figure 1. We recognize the importance of transparency in the selection process. To address this, we included details of the selection and exclusion criteria for each step in the review process.

    For point 3: Thank you for highlighting this issue regarding the terminology used in the text. We have revised the phrase to specify "25-hydroxyvitamin D concentration" instead of "vitamin D levels

    For point 4: Thank you for highlighting this issue. We have revised the phrase and made the necessary adjustments to enhance clarity.

    For point 5: Thank you for your thoughtful comment regarding the inclusion of in vitro and in vivo studies. We value your concerns about their relevance to assessing food fortification and vitamin D status in populations.

    These studies are integral to our analysis as they provide a comprehensive understanding of the factors influencing vitamin D fortification. Specifically, they are essential for evaluating the stability, bio-accessibility, and bioavailability of vitamin D in fortified foods.

    Stability ensures that vitamin D levels remain sufficient throughout the food's shelf life, which is critical for effective fortification. Bio-accessibility, defined as the proportion of vitamin D released from the food matrix in the gastrointestinal tract, is crucial for determining how much of the nutrient can potentially be absorbed by the body. Bioavailability further assesses the rate and extent to which vitamin D is absorbed and utilized at the cellular level, directly impacting population vitamin D status.

    By incorporating these aspects, we aim to provide a well-rounded perspective on the effectiveness of food fortification strategies, ultimately enhancing our understanding of how they can improve vitamin D levels within populations. We hope this clarifies the importance of including these studies in our review.

    For point 6: Thank you for your thorough review and for highlighting this error. You are correct that the description of the additional double bond and methyl group applies to vitamin D2, not vitamin D3. We have made the necessary corrections to ensure the accuracy of this information. Your feedback is greatly appreciated and contributes significantly to the clarity of our manuscript.

    Response to Comments on the Quality of English Language:Thank you for pointing this out. The manuscript has been revised by professional native English-speaking experts and adjustments have been made accordingly.

Reviewer 3 Report

Comments and Suggestions for Authors

References 1-9 are TOO OLD. Please search Google Scholar for articles and reviews regarding the health benefits of vitamin D and cite recent ones (past four years) with high numbers of citations adjusted for publication date. Health outcomes to include are cancer, cardiovascular disease, diabetes mellitus, all-cause mortality rate, pregnancy and birth outcomes, infectious diseases including COVID-19. Note that findings from observational studies are generally much stronger than those from vitamin D RCTs since most vitamin D RCTs have been based on guidelines for drugs, not nutrients:

Heaney RP. Guidelines for optimizing design and analysis of clinical studies of nutrient effects. Nutr Rev. 2014 Jan;72(1):48-54. doi: 10.1111/nure.12090.

Rostami M, Tehrani FR, Simbar M, Bidhendi Yarandi R, Minooee S, Hollis BW, Hosseinpanah F. Effectiveness of Prenatal Vitamin D Deficiency Screening and Treatment Program: A Stratified Randomized Field Trial. J Clin Endocrinol Metab. 2018 Aug 1;103(8):2936-2948. doi: 10.1210/jc.2018-00109.

Dawson-Hughes B, Staten MA, Knowler WC, Nelson J, Vickery EM, LeBlanc ES, Neff LM, Park J, Pittas AG; D2d Research Group. Intratrial Exposure to Vitamin D and New-Onset Diabetes Among Adults With Prediabetes: A Secondary Analysis From the Vitamin D and Type 2 Diabetes (D2d) Study. Diabetes Care. 2020 Dec;43(12):2916-2922. doi: 10.2337/dc20-1765.

Pilz S, Trummer C, Theiler-Schwetz V, Grübler MR, Verheyen ND, Odler B, Karras SN, Zittermann A, März W.

It would be very useful to summarize the findings from  the fortification studies for such  things as: 1, estimated fortification  (IU and µg per day)  per person for studies that provided such data; 2,  the amount of increase in 25(OH)D concentration that was/could be achieved by various vitamin D fortification approaches. Tables would be useful.

Also, a discussion should be made regarding how the increases in 25(OH)D might impact health outcomes. For this it is useful to find papers in the journal literature that give serum 25(OH)D concentration-health outcome relationships. Raising 25(OH)D above 50 nmol/L can be very useful for pregnancy , cardiovascular disease, and respiratory tract infection mortality rate.. While that would also reduce risk of cancer, higher concentrations would be better, but it would be very unlikely that any country would fortify food with enough vitamin D to go much higher.

A comprehensive litera- 12 ture search was conducted in PubMed and Google Scholar, focusing on studies from 2015 to 2024
Comment: No publications from 2024 were found in the reference list. Please extend the search through 2024.

Suggest citing some of these papers especially in the discussion section.

Rationale and plan for vitamin D food fortification: a review and guidance paper

Pilz, W März, KD CashmanME Kiely… - Frontiers in …, 2018 - frontiersin.org

Vitamin D food fortification and biofortification increases serum 25-hydroxyvitamin concentrations in adults and children: an updated and extended …

E DunlopME KielyAP JamesT SinghNM Pham… - The Journal of …, 2021 - Elsevier

Vitamin D and food fortification

KD CashmanM Kiely - Feldman and Pike's Vitamin D, 2024 - Elsevier

The effects of vitamin D-fortified foods on circulating 25 (OH) concentrations in adults: A systematic review and meta-analysis

B NikooyehTR Neyestani - British Journal of Nutrition, 2022 - cambridge.org

Vitamin D food fortification in European countries: the underused potential to prevent cancer deaths

T Niedermaier, T Gredner, S Kuznia… - European Journal of …, 2022 - Springer

The impact of vitamin D food fortification and health outcomes in children: a systematic review and meta-regression

R Al Khalifah, R Alsheikh, Y Alnasser, R Alsheikh… - Systematic reviews, 2020 - Springer

Trends in vitamin D status around the world

P Lips, RT de Jongh, NM van Schoor - JBMR plus, 2021 - Wiley Online Library

Vitamin D fortification of fluid milk products and their contribution to vitamin D intake and vitamin D status in observational studies—a review

ST Itkonen, M ErkkolaCJE Lamberg-Allardt - Nutrients, 2018 - mdpi.com

Safety of vitamin D food fortification and supplementation: evidence from randomized controlled trials and observational studies

FA Adebayo, ST Itkonen, T Öhman, M Kiely… - Foods, 2021 - mdpi.com

Summary outcomes of the ODIN project on food fortification for vitamin D deficiency prevention

M KielyKD Cashman - … Journal of Environmental Research and Public …, 2018 - mdpi.com

Vitamin D-fortified foods improve wintertime vitamin D status in women of Danish and Pakistani origin living in Denmark: a randomized controlled trial

IM GrønborgI TetensT Christensen… - European journal of …, 2020 – Springer

Vitamin D fortification of foods and prospective health outcomes

AN Moulas, M Vaiou - Journal of biotechnology, 2018 - Elsevier

One study evaluated the cost-effectiveness of vitamin D fortification, demonstrating

266

that fortifying milk and eggs with vitamin D is five times more cost-effective than pre-

267

scription drugs (43).

Comment: This paragraph needs more information. What were the prescription drugs? How about a comparison with vitamin D supplementation?

Author Response

Response and Revisions

Thank you for your evaluation of our article. We appreciate your feedback regarding its significance, organization, scientific soundness, references, and readability. We understand that there are areas for improvement, and your insights have been instrumental in guiding us. We are committed to enhancing the manuscript’s contribution to the field and improving its organization and clarity.

Comments 1:

References 1-9 are TOO OLD. Please search Google Scholar for articles and reviews regarding the health benefits of vitamin D and cite recent ones (past four years) with high numbers of citations adjusted for publication date. Health outcomes to include are cancer, cardiovascular disease, diabetes mellitus, all-cause mortality rate, pregnancy and birth outcomes, infectious diseases including COVID-19. Note that findings from observational studies are generally much stronger than those from vitamin D RCTs since most vitamin D RCTs have been based on guidelines for drugs, not nutrients:

  • Heaney RP. Guidelines for optimizing design and analysis of clinical studies of nutrient effects. Nutr Rev. 2014 Jan;72(1):48-54. doi: 10.1111/nure.12090.
  • Rostami M, Tehrani FR, Simbar M, Bidhendi Yarandi R, Minooee S, Hollis BW, Hosseinpanah F. Effectiveness of Prenatal Vitamin D Deficiency Screening and Treatment Program: A Stratified Randomized Field Trial. J Clin Endocrinol Metab. 2018 Aug 1;103(8):2936-2948. doi: 10.1210/jc.2018-00109.
  • Dawson-Hughes B, Staten MA, Knowler WC, Nelson J, Vickery EM, LeBlanc ES, Neff LM, Park J, Pittas AG; D2d Research Group. Intratrial Exposure to Vitamin D and New-Onset Diabetes Among Adults With Prediabetes: A Secondary Analysis From the Vitamin D and Type 2 Diabetes (D2d) Study. Diabetes Care. 2020 Dec;43(12):2916-2922. doi: 10.2337/dc20-1765. Pilz S, Trummer C, Theiler-Schwetz V, Grübler MR, Verheyen ND, Odler B, Karras SN, Zittermann A, März W.

Response 1: Thank you for your valuable feedback regarding the references. We appreciate your suggestion to include more recent articles and reviews on the health benefits of vitamin D. We have updated the reference list to include recent studies from the past four years that focus on health outcomes such as cancer, cardiovascular disease, diabetes mellitus, all-cause mortality, and infectious diseases, including COVID-19.

Comments 2:

It would be very useful to summarize the findings from the fortification studies for such things as:

  • 1, estimated fortification (IU and µg per day) per person for studies that provided such data;
  • 2, the amount of increase in 25(OH)D concentration that was/could be achieved by various vitamin D fortification approaches. Tables would be useful.

Response 2: Thank you for your valuable feedback regarding the summary of findings from the fortification studies. We appreciate your suggestion to highlight estimated fortification doses and the corresponding increases in serum 25(OH)D concentrations.

We would like to point out that our manuscript already includes detailed information on fortification doses in the relevant table, where we outline the fortification doses (in IU and µg) used across various studies. Additionally, we have included key findings that summarize the effects of these fortification strategies on serum 25(OH)D levels.

Comments 3:

Also, a discussion should be made regarding how the increases in 25(OH)D might impact health outcomes. For this it is useful to find papers in the journal literature that give serum 25(OH)D concentration-health outcome relationships. Raising 25(OH)D above 50 nmol/L can be very useful for pregnancy, cardiovascular disease, and respiratory tract infection mortality rate.. While that would also reduce risk of cancer, higher concentrations would be better, but it would be very unlikely that any country would fortify food with enough vitamin D to go much higher.

 Response 3: Thank you for your insightful comments. We appreciate your suggestion to explore the relationship between serum 25(OH)D levels and health outcomes. While we acknowledge the potential benefits of higher 25(OH)D concentrations for various health outcomes, our primary focus is to assess the impact and efficacy of vitamin D food fortification among apparently healthy adults only. However, we have included your recommendation in the discussion section, emphasizing the importance of future studies investigating how increases in serum 25(OH)D concentrations may influence health outcomes among populations at risk of deficiency, such as pregnant women, the elderly, and individuals with specific health conditions, including cancer, cardiovascular disease, and diabetes mellitus.

Comments 4:

A comprehensive litera- 12 ture search was conducted in PubMed and Google Scholar, focusing on studies from 2015 to 2024

Comment: No publications from 2024 were found in the reference list. Please extend the search through 2024.

 Response 4: Thank you for your valuable feedback. We appreciate your observation regarding the absence of publications from 2024 in the reference list. While we conducted a comprehensive literature search in PubMed and Google Scholar focusing on studies from 2015 to 2024, it is true that there were studies published in 2024. However, after careful screening based on our inclusion criteria, these studies were excluded because they were secondary studies (reviews) rather than primary research and their scope was not relevant to our targeted outcomes/ sample characteristics.

 Comments 5:

 Suggest citing some of these papers especially in the discussion section.

 Rationale and plan for vitamin D food fortification: a review and guidance paper S Pilz, W März, KD Cashman, ME Kiely… - Frontiers in …, 2018 - frontiersin.org

  • Vitamin D food fortification and biofortification increases serum 25-hydroxyvitamin D concentrations in adults and children: an updated and extended …E Dunlop, ME Kiely, AP James, T Singh, NM Pham… - The Journal of …, 2021 – Elsevier Vitamin D and food fortification KD Cashman, M Kiely - Feldman and Pike's Vitamin D, 2024 - Elsevier
  • The effects of vitamin D-fortified foods on circulating 25 (OH) D concentrations in adults: A systematic review and meta-analysis B Nikooyeh, TR Neyestani - British Journal of Nutrition, 2022 - cambridge.org
  • Vitamin D food fortification in European countries: the underused potential to prevent cancer deaths T Niedermaier, T Gredner, S Kuznia… - European Journal of …, 2022 - Springer
  • The impact of vitamin D food fortification and health outcomes in children: a systematic review and meta-regression R Al Khalifah, R Alsheikh, Y Alnasser, R Alsheikh… - Systematic reviews, 2020 - Springer
  • Trends in vitamin D status around the world P Lips, RT de Jongh, NM van Schoor - JBMR plus, 2021 - Wiley Online Library
  • Vitamin D fortification of fluid milk products and their contribution to vitamin D intake and vitamin D status in observational studies—a review ST Itkonen, M Erkkola, CJE Lamberg-Allardt - Nutrients, 2018 - mdpi.com
  • Safety of vitamin D food fortification and supplementation: evidence from randomized controlled trials and observational studies FA Adebayo, ST Itkonen, T Öhman, M Kiely… - Foods, 2021 - mdpi.com
  • Summary outcomes of the ODIN project on food fortification for vitamin D deficiency prevention M Kiely, KD Cashman - … Journal of Environmental Research and Public …, 2018 - mdpi.com
  • Vitamin D-fortified foods improve wintertime vitamin D status in women of Danish and Pakistani origin living in Denmark: a randomized controlled trial IM Grønborg, I Tetens, T Christensen… - European journal of …, 2020 – Springer
  • Vitamin D fortification of foods and prospective health outcomes AN Moulas, M Vaiou - Journal of biotechnology, 2018 - Elsevier

 Response 5: Thank you for your recommendations for citing relevant papers. We have carefully considered all the references you provided, including those related to the rationale for vitamin D food fortification, its effects on serum 25-hydroxyvitamin D concentrations, and associated health outcomes.

 Comments 6:  

 This paragraph needs more information. What were the prescription drugs? How about a comparison with vitamin D supplementation? [Line 266-267: “One study evaluated the cost-effectiveness of vitamin D fortification, demonstrating that fortifying milk and eggs with vitamin D is five times more cost-effective than prescription drugs (43).

Response 6: Thank you for your insightful comments regarding the need for more information in this section. To clarify, the study evaluated the cost-effectiveness of vitamin D fortification and indicated that fortifying milk and eggs with vitamin D is five times more cost-effective than the cheapest prescription drugs. Unfortunately, the specific prescription drugs used in the comparison were not detailed in the original paper. We have revised the text to reflect this clarification.

Round 2

Reviewer 2 Report

Comments and Suggestions for Authors

The authors have made a well considered effort to revise their manuscript in the light of this reviewer's comments. On the whole they have addressed the key issues raised. However the problem of the use of the term "vitamin D serum levels" has still not been fully corrected. Both vitamin D and 25-hydroxyvitamin D are present in serum. It is confusing to refer to vitamin D levels when what was measured was the concentration of 25-hydroxyvitamin D. This error is seen at the following lines in the mansucript: 177, 227, 278, 351, 365, 370.

Also at line 305, the word "doses" should read "does"

Author Response

Response and Revisions

Thank you for your thoughtful and constructive feedback. We appreciate your positive assessment. We value your input and are committed to making the necessary revisions to improve the manuscript.

Response 1: Thank you for your thoughtful feedback and for recognizing our efforts to address the comments provided. We appreciate your careful review and your continued attention to detail. Regarding your concern about the use of the term "vitamin D serum levels," we have now made the necessary corrections to ensure clarity. Specifically, we have replaced "vitamin D" with "25-hydroxyvitamin D" in the relevant sections of the manuscript

 Response 2: Thank you for your valuable feedback. I appreciate your careful review of the manuscript. I have revised the sentence to improve clarity. 

Reviewer 3 Report

Comments and Suggestions for Authors

The manuscript is improved.
There are a few erros such as misspelling of yogurt

and missing inforation for the final cited reference.

Author Response

Response and Revisions

Thank you for your detailed and constructive feedback on our manuscript. We greatly appreciate the time and effort you dedicated to reviewing our work. In response to your suggestions, we have carefully revised the manuscript to address the points you raised. We believe these changes have enhanced the clarity and quality of the work. Thank you again for your valuable input.

Comments 1:

The manuscript is improved. There are a few errors such as misspelling of yogurt and missing information for the final cited reference.

Response 1: Thank you for your valuable feedback on the manuscript. We are pleased to hear that you find the manuscript improved. We sincerely appreciate your careful review. Regarding the errors you pointed out, we have corrected the misspelling of "yogurt" as well as addressed the missing information in the final cited reference. We have thoroughly checked the manuscript to ensure all necessary details are now included.